# Eukaryotic Diversity Based on High-Throughput 18S rRNA Sequencing and Its Relationship with Environmental Factors in a Salt Lake in Tibet, China

**Lele He** [1,2]**, Qi Wang** [1,2]**, Zhe Wang** [1,2]**, Fang Wang** [3] **, Shichun Sun** [2,3] **and Xiaoshou Liu** [1,2,*]

1    College of Marine Life Sciences, Ocean University of China, Qingdao 266003, China
2    Institute of Evolution and Marine Biodiversity, Ocean University of China, Qingdao 266003, China
3    Fisheries College, Ocean University of China, Qingdao 266003, China
*    Correspondence: liuxs@ouc.edu.cn; Tel.: +86-532-82031735

**Abstract:** Eukaryotes exist widely in aquatic ecosystems. It is of great importance to study their species composition, diversity, and relationship with environmental factors to protect and maintain ecosystem balance. Salt lakes are essential lakes rich in biological and mineral resources and have significant research value. To understand the characteristics of eukaryotic diversity in salt lake sediments, we conducted a sampling survey of the benthos in Kyêbxang Co, Tibet, in July and August 2020. The sampling area was divided into littoral, sublittoral, and profundal zones. A total of 42 species of Metazoa, 159 species of Protozoa, 63 species of Viridiplantae, and 46 species of Fungi were identified by the high-throughput sequencing of 18S ribosomes. Alpha diversity analysis revealed significant differences in species composition among the three study zones. The littoral zone had the highest Sobs index and Chao index, indicating that the eukaryotic diversity and richness in this zone were significantly higher than those in the profundal and sublittoral zones. Redundancy analysis (RDA) showed that water depth, temperature, and sediment organic matter content significantly affected the community structure of eukaryotes zones, especially the distribution of dominant genera such as *Dunaliella*, *Psilotricha* and *Brachionus*. Cooccurrence network analysis showed that *Dunaliella*, *Aphelidium*, temperature, water depth, and organic matter represent essential nodes in the entire network. This study can provide baseline data and new insights for eukaryotic diversity research for salt lakes.

**Keywords:** salt lakes; high throughput sequencing; diversity; cooccurrence network; correlation analysis; kyêbxang co; tibet





## 1. Introduction

Salt lakes usually refer to lakes with salt content $\geq 3.0$ g/L as an important lake type with abundant mineral and biological resources. Ecosystems of saline lakes are sensitive to the impact of external factors. Changes in weather conditions often lead to fluctuations of water salinity, which may cause a changeover in the hydro-biological regime of the lake [1–3]. The various organisms in salt lakes play an important role in increasing the diversity and maintaining the balance of salt lake ecosystems [4]. Benthic eukaryotes can modify sediment habitats [5–7], and their participation in the formation of benthic food webs makes an essential contribution to the material cycling and energy transfer of aquatic ecosystems [8,9]. In addition, benthic eukaryotic communities are susceptible to changes in external environmental conditions [10]. The changes in their communities can objectively reflect the changes in environmental quality, so they are one of the critical groups in biological monitoring [6,11]. Research shows that the diversity present in the egg bank of saline lakes is higher than that recorded in the water column at any time [12]. It indicates that in each period, the water column shows only a portion of the biodiversity that the sediment contains unexpressed as resting stages. This could

provide complementary suggestions for understand the space-time distribution of plankton organisms [13]. Studying the response pattern of eukaryotic communities in salt lake sediments to environmental factors and revealing the links between their structure and function will be essential for further research on the ecological environment of salt lakes [14]. In the past, traditional morphological identification methods were used to describe the characteristics and distribution of organisms, and there was a lack of research on the relationship between organisms and environmental factors. This morphological method is not only affected by sampling conditions and preservation techniques but is also subject to significant variation and disagreements in identifying these organisms [15,16]. The environmental DNA (eDNA) technique provides a new approach for investigating benthic eukaryotic communities [17,18]. We can extract environmental DNA and perform high-throughput sequencing to get information about species composition and abundance in the environment. Based on the characteristics of a large amount of sequencing data, high accuracy, high sensitivity, and low cost, high-throughput sequencing technology has been applied to studying eukaryotic communities in various environments [19].

Salt lakes in China are mainly distributed on the Qinghai-Tibet Plateau [20]. Due to the high salt content of salt lakes, only specific eukaryotic taxa can survive there [21]. Most salt lakes are remote, the natural environment is harsh, the research conditions are not convenient, and only a few studies have been conducted on Tibetan salt lakes [22,23]. Therefore, comprehensive and detailed data on biologic community structure in most salt lakes have been lacking [24]. Kyêbxang Co is a significant salt lake on the Tibetan Plateau, and many rivers flow into it, and it is rich in brine shrimp (*Artemia* spp.) resources [25]. Kyêbxang Co has a salinity of over 44‰ and is a mesosaline lake [26]. The brine shrimp resources in salt lakes are vital for aquaculture feeds, so the ecological environment of Kyêbxang Co is closely related to the income of millions of herders. Through high-throughput sequencing, the structure and composition of the eukaryotic community in Kyêbxang Co were investigated, emphasizing the analysis and exploration of eukaryotic community characteristics and their relationships with the environment. This study aimed to provide more knowledge about the eukaryotic resources and ecological environment of the Tibetan salt lakes and to provide a scientific basis for further research, development, and protection of the salt lake.

## 2. Materials and Methods

### 2.1. Sample Collection and Physicochemical Factor Determination

In July 2020, the field survey was carried out in Kyêbxang Co of Tibet (Figure 1), located in the north of Tibet and the hinterland of Qinghai-Tibet Plateau with an average elevation of more than 4500 m. The sampling area was divided into littoral, sublittoral, and profundal zones. There were three sites (sampling sites), B4, B2, and H6, in the littoral zone, located at the three corners of Kyêbxang Co, respectively, and the water depth was less than 9 m. There were two sites in the sublittoral zone, H4 and F2, far from the shore and they had a deeper water depth (9–16 m) than the littoral zone. Four sites (C3, D2, D4, F4) were in the profundal zone, at the centre of the lake, with a water depth exceeding 16 m. The specific information for each site is shown in Table 1. The sediment samples were collected with a grab dredge, and the in situ physicochemical factors were determined. After collection, the samples were immediately put into sterile polyethene sealed bags and placed on ice to be rapidly transported back to the laboratory. After returning to the laboratory, each sample was divided into two parts: one was stored at 4 °C to determine physicochemical indices; the other was saved in a −80 °C freezer in sterile centrifuge tubes for eukaryotic DNA extraction. According to the methods described in the literature, the water temperature (T), pH, and dissolved oxygen (DO) at each sampling site were measured with a YSI 6920 sonde in the field. Sediment environmental factors were determined in the laboratory, including organic carbon (OM), chlorophyll a (Chla), water content (MC), and grain size, and the elemental analyser was used to determine the content of C, N, and S elements. All

operations were performed by the "Water quality. Guidance on sampling techniques from lakes, natural and man-made" (GB / T 14581-1993).

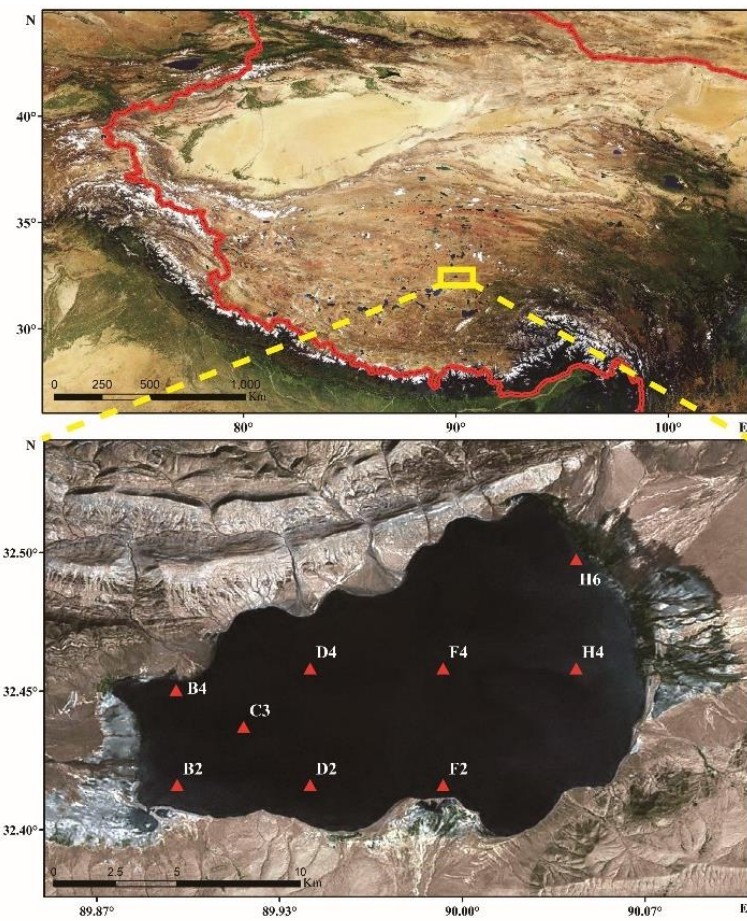

**Figure 1.** Map of the locations of the sampling sites in Kyêbxang Co, Tibet, China.

**Table 1.** Environmental factors of each sampling site in Kyêbxang Co, Tibet, China.

|  | B2 | B4 | C3 | D2 | D4 | F2 | F4 | H4 | H6 |
|---|---|---|---|---|---|---|---|---|---|
| WD (m) | 7 | 1.5 | 16.3 | 16.2 | 23 | 12.8 | 21.5 | 9.5 | 5.4 |
| T (°C) | 13.76 | 15.29 | 1.07 | 1.25 | 0.95 | 3.85 | 0.73 | 13.04 | 13.81 |
| pH | 8.95 | 8.84 | 8.81 | 9 | 8.88 | 9.07 | 8.92 | 9.04 | 9 |
| DO (mg/L) | 4.84 | 6.06 | 1.19 | 6.08 | 0.08 | 11.7 | 9.71 | 2.06 | 4.67 |
| MC (%) | 46.32 | 36.87 | 46.42 | 51.48 | 49.64 | 52.99 | 60.64 | 44.11 | 40.72 |
| OM (%) | 2.86 | 0.43 | 3.58 | 7.8 | 7.47 | 4.51 | 8.6 | 1.93 | 2.83 |
| Chla (ug/L) | 0.61 | 1.8 | 0.56 | 0.47 | 0.4 | 0.34 | 0.6 | 0.89 | 0.71 |
| N (%) | 0.09 | 0.08 | 0.26 | 0.27 | 0.2 | 0.13 | 0.37 | 0.11 | 0.14 |
| C (%) | 4.84 | 1.89 | 5.73 | 8.89 | 8.67 | 6.29 | 10.04 | 3.84 | 3.98 |
| S (%) | 0.35 | 0.38 | 0.87 | 1.29 | 0.73 | 0.36 | 1.24 | 0.38 | 0.73 |
| C/N | 53.78 | 23.63 | 22.04 | 32.93 | 43.35 | 48.38 | 27.14 | 34.91 | 28.43 |
| Sediment type | Sand | Silty sand | Silty sand | Silty sand | Silty sand | Silty sand | Silty sand | Silty sand | Silty sand |

Note: WD: Water depth; T: water temperature; DO: dissolved oxygen; MC: water content; OM: organic carbon; Chla: chlorophyll a; N: nitrogen content; C: carbon content; S: sulphur content; C/N: ratio of carbon content to nitrogen content.

## 2.2. DNA Extraction and Sequencing of Benthic Eukaryotes

Take the sediment sample out of the −80 °C refrigerator and put it on ice to keep the mud sample low temperature. Use the power soil DNA isolation Kit (MOBIO) and refer to the instructions to extract the total DNA in the sample. To determine concentration and purity, the extracted DNA was analysed with a micro-ultraviolet spectrophotometer



(NanoDrop ND-1000, Wilmington, DE, USA). Nine samples were sequenced by the Illumina miseq platform in Shanghai Meiji Biomedical Technology Co., Ltd. The V4 region of the eukaryotic 18S rRNA gene was amplified using the general primers TAReuk454FWD1F (5′-CCAGCASCYGCGGTAATTCC-3′) and TAReukREV3R (5′-ACTTTCGTTCTTGATYRA-3′) to obtain the target DNA sequences from the different samples [27]. The PCR was carried out using the TransGenAP221-02: TransStart FastPfu DNA Polymerase, 20 μL reaction system. The reaction conditions for PCR amplification were 95 °C, 3 min, 35× (95 °C, 30 s; 55 °C, 30 s; 72 °C, 45 s); 72 °C, 10 min, and 10 °C until stopped. The PCR amplification system included 5×FastPfu Buffer (4 μL), 2.5 mM dNTPs (2 μL), five μM forward primer (0.8 μL), five μM reverse primer (0.8 μL), FastPfu Polymerase (0.4 μL), BSA (0.2 μL), and DNA template (10 ng), with ddH$_2$O added to attain a total volume of 20 μL. The PCR amplification products of the different samples were sequenced by Shanghai Majorbio Bio-pharm Technology Co., Ltd.

### 2.3. Data Analysis and Processing

The 18S V4 region was MiSeq amplified, raw data obtained from the sequencing procedure were subjected to splicing and quality control, and chimeras were removed to obtain optimized sequences. The OTU clustering was performed based on comparing the optimized sequences with the Silva 128/18S eukaryotic database, and an OTU abundance table was constructed for subsequent analysis. According to sequence similarity, the RDP classifier algorithm was used for the taxonomic analysis of OTU with a 97% similarity level. The community composition of each sample was evaluated at each taxonomic level. After the original OTU table was obtained, the OTU table was drawn according to the minimum sequence number of samples to avoid the error of subsequent analysis results caused by different sequencing depths. In addition, the software mothur was used to calculate Chao1, ACE, Simpson, Shannon, and coverage index values for Alpha diversity analysis [28,29]. Differences in the community structure of eukaryotes among different samples were compared and analysed using STAMP software ($p < 0.05$). The UPGMA was used for eukaryotic community distribution and cluster analysis. Principal co-ordinates analysis (PCoA) and correlation analysis were used to evaluate the community distribution and perform principal component and cluster analysis on the eukaryotes.

The analysis of the relationships between eukaryotic community diversity and environmental factors and the one-way ANOVA of environmental factors among the different sampling sites were completed in SPSS 22.0 ($p < 0.05$). The relevant figures were drawn with Excel and Origin9.0. Redundancy analysis (RDA) was completed in Canoco5. Network analysis was completed in Cytoscape v3.7.1. (taxonomic abundance ≥50, Spearman correlation coefficient >0.5 and $p < 0.05$).

Sequences generated from the samples used in this study have been deposited in the Sequence Read Archive (SRA) database of the NCBI. The accession number for the 9 samples collected from the Kyêbxang Co is PRJNA818805.

## 3. Results

### 3.1. Analysis of Environmental Factors

The physicochemical factors of the sediments are the main environmental factors correlated with the distribution of eukaryotes. The overall characteristics of the leading environmental factors of the nine sites are shown in Table 1. The water depth in the littoral zone was 1.5–7 m, with an average of 4.6 m. The water depth of the sublittoral zone was 9.5–12.8 m, with an average of 11.2 m. The water depth in the profundal zone was 16.2–23 m, and the average water depth was 19.3 m. There was a noticeable depth gradient between the three zones. In addition to the water depth, the water temperature of the sediment also showed prominent gradient characteristics.

At sites B4, B2, H6, and H4, with a water depth of less than 10 m, the water temperature was about 13 °C, while when the water depth was more than 10 m, the water temperature dropped sharply to less than 4 °C. Therefore, it was indicated that there may

be a thermocline in Kyêbxang Co, which has a water depth of about 10 m. More detailed investigation is needed to confirm this phenomenon and whether this conclusion can be extended to other salt lakes in Tibet. However, we know that the temperature (T) variable will have a certain impact on the community structure of eukaryotes, which will be discussed in the next part of our research. The pH value of 9 sites in Kyêbxang Co fluctuated slightly, with the lowest of 8.81, the highest of 9.07, and the average of 8.95, indicating that the water quality in Kyêbxang Co is alkaline. The DO of the nine sites in Kyêbxang Co was not distributed according to the law of water depth like the T. The peak appeared in the F2 site, while the water depth of the F2 site was 12.8 m, which was inconsistent with the general situation, probably because there was an undercurrent surge at the middle and bottom of Kyêbxang Co, but it seems to contradict the existence of thermocline. The specific explanation needs a more detailed investigation. In addition, the DO of bottom water was greater than the DO of surface water at many survey points such as F2. The MC of sediment also showed no obvious trend. The highest value of OM was 8.6% at the F4 site, and the lowest was 0.43% at the B4 site. This may be because the water column is shallower in the most coastal site, and the planktonic community does not produce the same organic matter (sinking to the bottom) which could be produced in the highest water column elsewhere. The content of Chla in the sediment can represent the magnitude of primary productivity, and the maximum value was 1.8μg/g at site B4, which may be because there are more photosynthetic producers in the shallow water with sufficient light. The elemental analyser was used to determine the contents of C, N, and S in sediments. The results showed that the highest C content in sediments was at F4, which was 10.04%, and the lowest was at B4, which was 1.89%. It was consistent with the trend of OM; the two points with the highest content of S were F4 and D2, which were 1.24% and 1.29%, respectively. This may be because F4 and D2 belong to the profundal zone with profundal depth and high sulphide content. The sediment types of the nine sites were primarily silty sand, and only the B2 site was sand.

### 3.2. Analysis of Eukaryotic Community Structure in Kyêbxang Co

3.2.1. Sequencing Statistics, Eukaryotic Species Composition

After sequencing, 355,567 sequences were obtained from nine samples, which were divided into 559 OTUs based on a 97% similarity level. Then the samples were diluted according to the minimum number of sample sequences. A total of 286,920 sequences and 524 OTUs were obtained from all samples, and 208 species belonging to 37 phyla, 77 classes, 120 orders, 144 families, and 167 genera were annotated by taxonomy. The method of random sampling of sequences using mothur software is used to construct the dilution curve with the number of extracted sequences and their Sobs index, as shown in Figure 2. The results showed that the rarefaction curves gradually became stable as the number of sequencing reads increased, indicating that the amount of sequencing data was reasonable, and the experimental results reflected the species diversity of the samples.

According to the Venn diagram (Figure 3) of the three locations, there were 193 unique species in the littoral zone, accounting for 36.83% of the total OTUs. However, there were only 35 and 37 unique species in the profundal and sublittoral zones, accounting for 6.78% and 7.06% of the total OTUs, respectively. It showed more endemic species in the littoral zone, while fewer endemic species were in the sublittoral and profundal zones. Moreover, 110 species were simultaneously found in all three areas, accounting for 20.99% of the total OTUs. A total of 68 species were found in littoral and sublittoral zones, 55 species in littoral and profundal zones, and 26 species in sublittoral and profundal zones, indicating noticeable species differences among the three areas more eukaryotes live in littorals. This may be due to the shallower water depth and higher DO in sediments in the littoral zone, while most eukaryotes are aerobic and thus have higher abundance. In addition, from the perspective of the whole trend, 426 OTUs were obtained from the sequencing results of the littoral zone, which is nearly twice that of the sublittoral (241 OTUs) and the profundal

(226 OTUs) zones, which is also sufficient to show that the biological types in the littoral zone are more abundant.

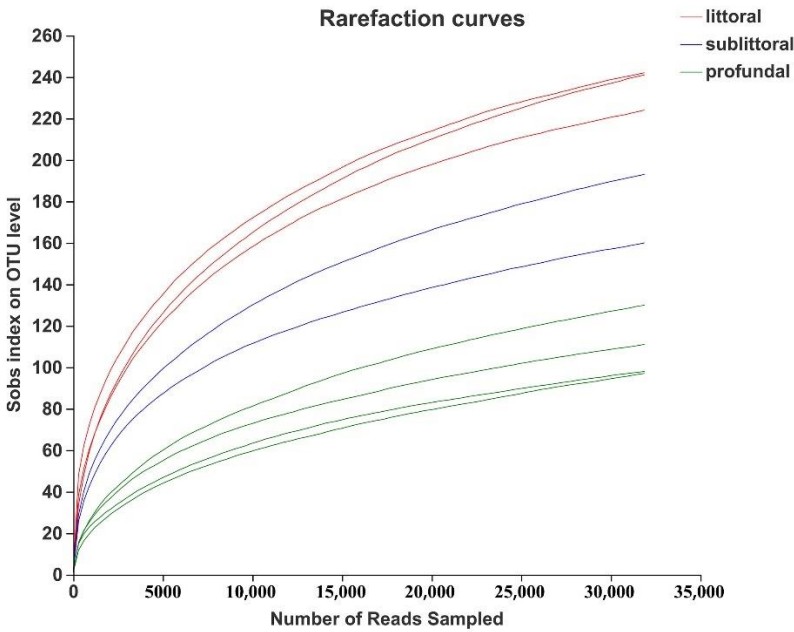

**Figure 2.** Rarefaction curves of eukaryotes at the sampling sites in Kyêbxang Co, Tibet, China.

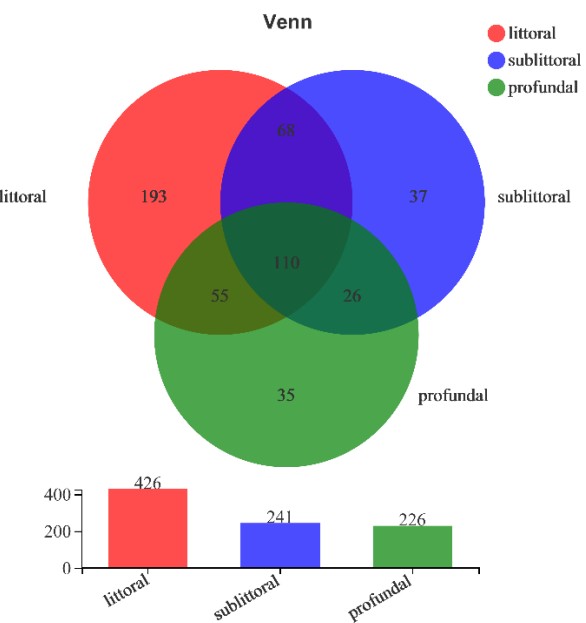

**Figure 3.** Venn map of eukaryotic communities in different regions of Kyêbxang Co, Tibet, China (OTU level).

### 3.2.2. Analysis of the Alpha Diversity of Eukaryotes

The Sobs index refers to the number of OTUs observed, while the Shannon index comprehensively considers the richness and evenness of the community. The higher the Shannon index was, the higher the community diversity was. The higher the Simpson index, the higher the community dominance; the more prominent the Chao1 or ACE index, the higher the richness of the community; the coverage index indicates the coverage of sample species sequencing. Alpha diversity was analysed for nine stations in three regions in this study, and the specific Alpha diversity indices for each station are listed in Table 2. The average Alpha diversity indices for the three regions are presented in Table 3, and

the Sobs and Chao indices, which were significantly different, are plotted in Figure 4. The results show that the average Sobs index in the littoral zone was 235.67, which was significantly higher than 176.50 in the sublittoral zone and even more than twice that in the profundal zone 109, indicating that the number of OTUs in the littoral zone was the largest. The highest Sobs and Shannon indices were found at the H6 site, indicating high biomass and biodiversity. Moreover, the average Shannon index (2.63) of the littoral zone was much higher than that of the sublittoral (1.83) and profundal (1.43) zones, indicating that the diversity of eukaryotes in the sample was the highest in the littoral zone and the lowest in the profundal zone. In addition, the Chao index of the littoral zone was significantly higher than that of the profundal zone, which indicates that the species richness was the highest in the littoral zone and the lowest in the profundal zone. The results of the Kruskal–Wallis test on the Sobs index and the Chao index showed significant differences among the three study zones. The coverage index of all sites was more than 99%, indicating that sequencing had good coverage of species.

**Table 2.** Alpha diversity indices of eukaryotes in sediments of different sites in Kyêbxang Co, Tibet, China.

| Sample | Sobs | Shannon | Simpson | Ace | Chao | Coverage |
|--------|------|---------|---------|-----|------|----------|
| C3 | 130 | 1.6340 | 0.2785 | 220.3709 | 181 | 0.9985 |
| F4 | 98 | 1.5741 | 0.2907 | 135.2014 | 133 | 0.9989 |
| D2 | 97 | 1.2520 | 0.4156 | 193.8154 | 179 | 0.9987 |
| F2 | 160 | 1.8418 | 0.2993 | 256.5098 | 231 | 0.9984 |
| B2 | 224 | 2.2485 | 0.3082 | 267.1607 | 266 | 0.9983 |
| H4 | 193 | 1.7892 | 0.3772 | 257.9494 | 259 | 0.9981 |
| D4 | 111 | 1.2460 | 0.4458 | 196.7759 | 160 | 0.9988 |
| H6 | 242 | 3.3565 | 0.0587 | 292.8188 | 281 | 0.9982 |
| B4 | 241 | 2.2952 | 0.2007 | 309.3496 | 304 | 0.9979 |

**Table 3.** Average values of Alpha diversity indices of eukaryotes in three regions of Kyêbxang Co, Tibet, China.

| Zone | Sobs | Shannon | Simpson | Ace | Chao | Coverage |
|------|------|---------|---------|-----|------|----------|
| littoral zone | 235.6667 ± 8.2597 | 2.6334 ± 0.5117 | 0.1892 ± 0.1022 | 289.7764 ± 17.3574 | 283.5118 ± 15.8694 | 0.9981 ± 0.0002 |
| sublittoral zone | 176.5000 ± 16.5000 | 1.8155 ± 0.0263 | 0.3383 ± 0.0390 | 257.2296 ± 0.7198 | 244.6945 ± 13.8611 | 0.9983 ± 0.0001 |
| profundal zone | 109.0000 ± 13.3229 | 1.4265 ± 0.1788 | 0.3577 ± 0.0740 | 186.5409 ± 31.3763 | 163.4338 ± 19.3142 | 0.9987 ± 0.0002 |

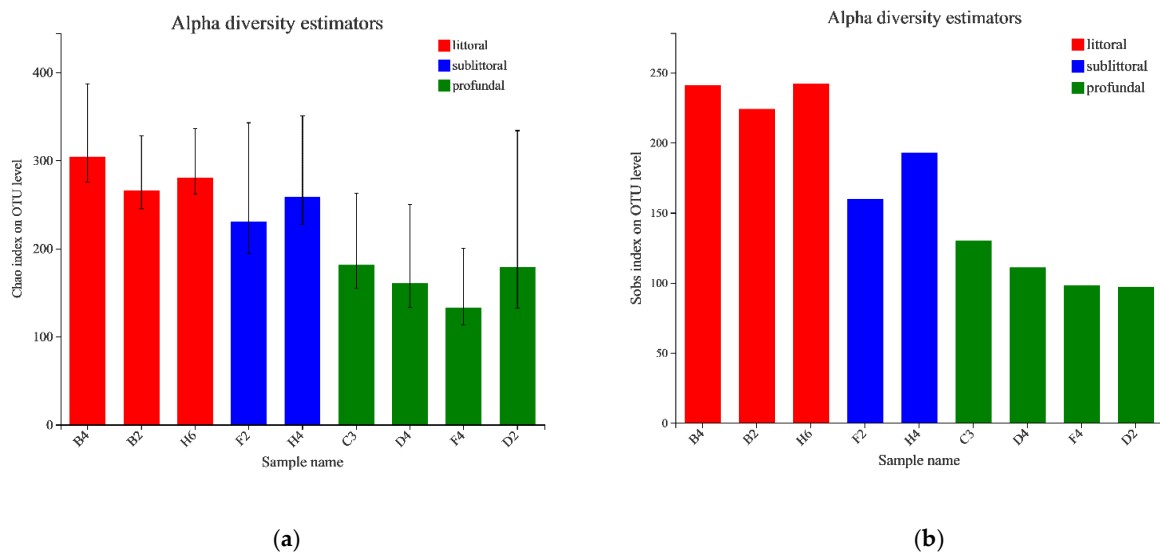

(a)      (b)

**Figure 4.** Alpha diversity assessment of eukaryotes in Kyêbxang Co, Tibet, China. Chao (**a**) index was used to represent community taxa richness and Sobs (**b**) was the observed OTU number.

### 3.2.3. Analysis of the Beta Diversity of Eukaryotes

To investigate the similarity of eukaryotic biodiversity among samples, PCoA (principal co-ordinates analysis) analysis (Figure 5) was conducted in this paper. The distance and similarity between samples can be reflected by analysing the OTU composition of different samples. The results of PCoA analysis show that, as shown in Figure 5, coordinate axis PCo1 could explain 55.82% of the differences in eukaryotic community composition, and coordinate axis PCo2 could explain 17.53% of the changes in the eukaryotic community. In the PCo1 dimension, the littoral, sublittoral, and profundal zones are very different, and especially the littoral zone is separated from the other two areas. From the dimension of PCo2, the littoral and profundal zones are basically at the same level, while the sublittoral zone is different from the first two, located below the coordinate axis. In terms of geographical location, each site in the littoral zone was located at the edge of the lake in different directions. However, it can be accurately clustered in the same area in the PCoA diagram, which also shows that the eukaryotic community structure in the littoral zone was very similar and the rationality of our grouping scheme.

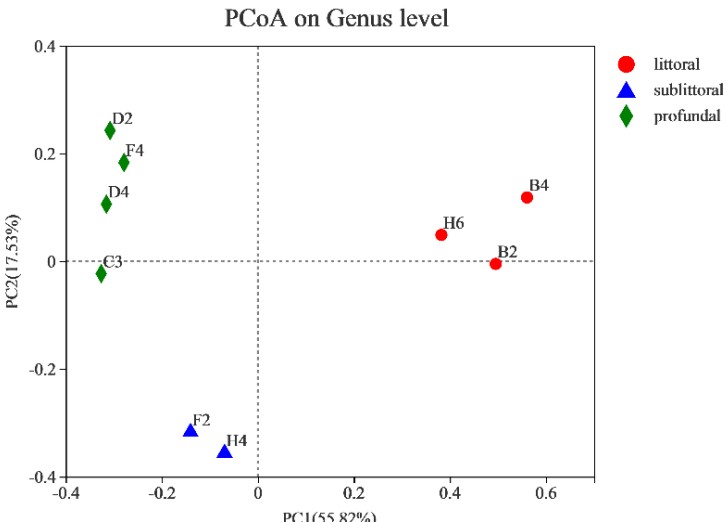

**Figure 5.** PCoA analysis of eukaryotic community diversity in Kyêbxang Co, Tibet, China.

### 3.2.4. Taxonomic Composition and Difference Analysis of Eukaryotes

The relative abundance distribution of eukaryotes at the phylum level in sediments from nine sites in Kyêbxang Co is shown in Figure 6. By comparison, it can be found that Chlorophyta, unclassified Eukaryota, Rotifera, Ciliophora, Arthropoda, Ascomycota, Haptista, Cercozoa, Nematoda, and Basidiomycota were the taxa in the top ten of abundance. The unclassified Eukaryota (red) were overwhelmingly dominant in the littoral zone, accounting for an average of 50.02%, more than half of the total, while their proportion drops sharply in the sublittoral and profundal zones. In addition, the proportion of Ciliophora (sky blue) was also high in the littoral zone, with slight differences between the three sites along the littoral zone, with the highest proportion of Ciliophora reaching 40.10% in the B4 site. In contrast, the highest proportion of unclassified Eukaryota reached 67.23% in the B2 site, with relatively few Ciliophora (8.11%). Many Rotifera (13.84%) were present in the H6 site. In contrast, Rotifera were found in high numbers in the sublittoral and profundal zones, indicating that the H6 site may be at the transitional position between littoral and profundal zones. In conclusion, although there are subtle differences in different sampling sites along the littoral zone, the general trend is the same.

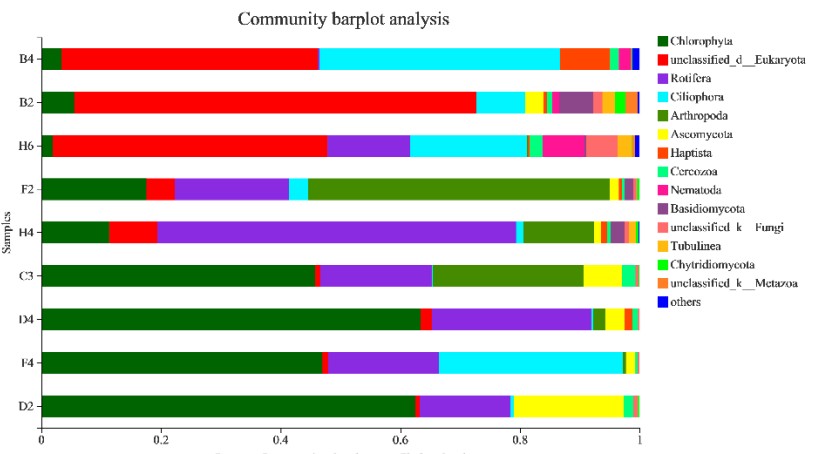

**Figure 6.** Taxa composition of eukaryotes at each site (phylum level) in Kyêbxang Co, Tibet, China.

The dominant species in the sublittoral zone were Rotifera (purple) and arthropods (light green). The two sites in the sublittoral zone focused on each, with the F2 site arthropod share reaching 50.37% compared with only 19.09% for Rotifera. The H4 site arthropod share was 59.98%, compared with only 11.73% for arthropods. It is worth noting that a certain number of Chlorophyta (dark green) appeared at the F2 and H4 sites in the sublittoral zone, accounting for 17.56% and 11.32%, respectively.

In the profundal zone, there were many Chlorophyta (green) and Rotifera (purple), of which the average proportion of Chlorophyta reached 54.67%, more than half, indicating that Chlorophyta was the absolute dominant taxon in the profundal zone. In addition, Rotifera had a large proportion of distribution in the four sites in the profundal zone, with an average of 19.76%. However, the sites in these four profundal zones have their characteristics. For example, there are many Arthropoda (25.25%) in the C3 site and a large number of Ciliophora (30.83%) in the F4 site. It is unclear why these Ciliophora, dominant taxa in the littoral zone, appear in the F4 site in the profundal zone. In addition, the proportion of Ascomycota (yellow) in the profundal zone began to increase and even surpassed the Rotifera at the D2 site, reaching 18.38%.

The sample and species relationship diagram (Figure 7) shows the distribution proportion of dominant taxa in each sample and the distribution proportion of dominant taxa in different samples through the visual circle diagram.

In the Circos plot, the left half circle represents the taxa composition for the sample. Different colours in the outer colour bands represent different groupings, the inner colour bands represent different taxa, and the length represents the relative abundance of that taxa in the corresponding sample. The right half-circle indicates the distribution proportion of taxa in different sites at that taxonomic level. The outer colour bands represent taxa, the inner colour bands represent different groupings, and the length represents the distribution proportion for that sample in a given taxon. Analysis of the composition of the eukaryotic community at the phylum level Circos plots (Figure 7) revealed that the eukaryotic taxa for all samples included: Chlorophyta, unclassified Eukaryota, Rotifera, Ciliophora, Arthropoda, Ascomycota. The taxon with the highest species richness was Chlorophyta, which was mainly distributed in the four sites D2, D4, F4, and C3 at profundal zone depth, but less frequently at the other sites with 7.94%. Whereas the H4 site had many Rotifera accounting for up to 60%, these results are consistent with the Heatmap representation results above. However, from the Circos plot, we can also find other exciting phenomena from different perspectives. The Ciliophora were mainly distributed in the B4 (39%) and F4 (30%) sites, and the occupation ratio of other sites was less. The environmental factors at these two sites both presented apparent differences. For example, the water depth was 1.5 m for B4 compared with 21.5 m for F4. Our study has not found a plausible explanation concerning what caused so many Ciliophora in these two sites, with vastly different geo-

graphic and physicochemical factors. In addition, we note that the Arthropoda were mainly derived from the F2 (56%), C3 (28%), and H4 (13%) sites. In contrast, the contribution values for other too deep or shallow sites are almost zero, indicating that the Arthropoda prefer to live in specific aqueous environments.

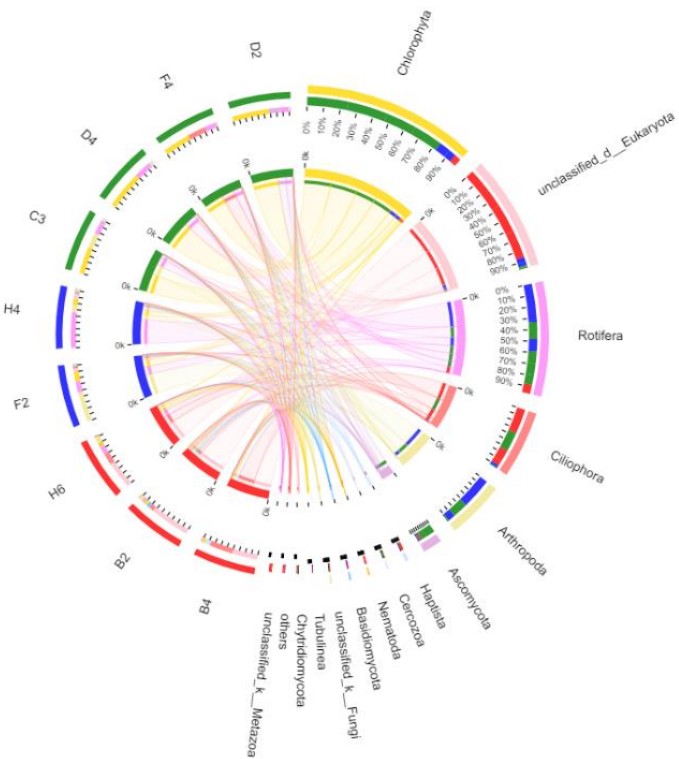

**Figure 7.** Samples correspond to the relative abundance of eukaryote sequences at the phylum level in Kyêbxang Co, Tibet, China. In the Circos diagram, the small semicircle on the left represents the phylum composition of the sample, the colour of the outer band represents the sampling site, the colour of the inner band represents the phylum, and the length represents the relative abundance of the phylum in the corresponding sample. The large semicircle on the right represents the proportional distribution of each phylum in the different samples, the outer band represents the phyla, the inner band represents the different sites, and the length represents the proportional distribution of the sequences of each phylum that were from each sample.

To observe the differences in the eukaryote community structure in different regions, samples from the three regions were selected based on community abundance data using rigorous statistical methods for the Kruskal–Wallis rank-sum test to assess the significance of observed differences. At the genus level, organisms were ranked in order of abundance from top to bottom (Figure 8). According to the different test results, it can be found that in the three most abundant taxa: Dunaliellaceae, *Brachionus*, and unclassified Eukaryota, the differences among the three studied regions were significant ($p < 0.05$), which indicated that different dominant species survived in different habitats. *Dunaliella* increased in littoral, sublittoral, and profundal zones, while unclassified Eukaryota decreased gradually in these three regions. In comparison, *Brachionus* was more abundant in the intermediate sublittoral zone and less abundant in the littoral and profundal zones, similar to the distribution of Arthropoda. In addition, other taxa with higher abundances, such as *Psilotricha* and *Saitozyma*, were significantly different between the three studied regions.

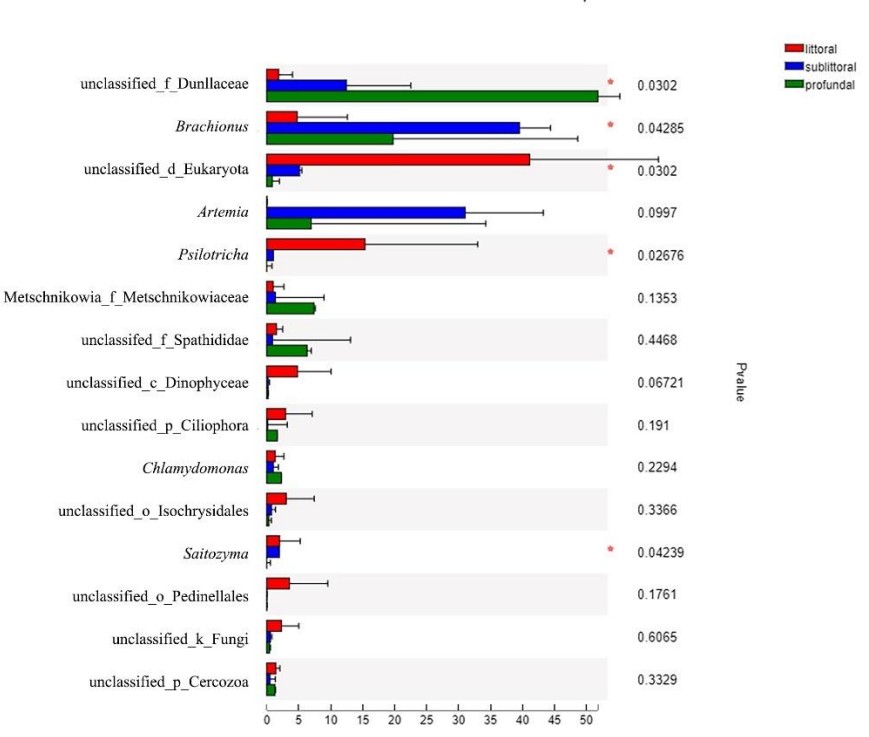

**Figure 8.** Analysis of significant differences for the dominant taxa among the littoral, sublittoral and pro-fundal areas in Kyêbxang Co, Tibet, China. Note: * indicates significant differences among the three areas at 0.05 level.

The top 30 taxa in abundance were subjected to cluster analysis, and a heat map was drawn (Figure 9), with red representing more abundant taxa and white representing less abundant taxa. The overall results show differences in the dominant taxa in the sediment environment at different sites. The unclassified Eukaryota accounted for a higher proportion of the three sites along the littoral zone and a lower proportion of the other sites. However, *Brachionus* was more abundant at the H6 site along the littoral zone, sublittoral and profundal zones. On the other hand, *Dunaliella* appears to occur abundantly only in the sublittoral and profundal zones. *Artemia* was found abundantly at both sites in the sublittoral zone and less frequently along the littoral or profundal zones, suggesting that a water depth of 9–16 m in the sublittoral zone may be well suited for its survival.

From the cluster analysis (Figure 9), the three sites, B4, B2, and H6, along with the littoral zone cluster into one category, while the other sites can be further divided into two main categories, the profundal zone (F4, C3, D4, D2) and the sublittoral zone (F2, H4), respectively. The littoral zone B2 and H6 sites were relatively closer to the B4 site in the cluster analysis, probably because the water depths of the B2 and H6 sites were similar, 7 m and 5.4 m, respectively. The water depth at the B4 site was only 1.5 m, illustrating the critical influence of water depth on eukaryotic communities. Whereas the two nearest to the four sites in the profundal zone were the D4 and D2 sites, followed by C3 and then F4, the geographically closest C3 and D4 sites did not cluster together in the cluster analysis, and it was seen that the geographic location was not the only factor affecting the community structure. Meanwhile, the sublittoral and profundal zones clustered into the same cluster in cluster analysis, while the littoral zone was a separate cluster. The habitats in the centre of the lake are distinct from those in the littoral zone, but can be further divided into the sublittoral and profundal zones. This illustrates that the regional habitats in the centre of the lake were also not wholly congruent. The physicochemical factors of the habitat change again when a critical depth is reached, leading to the emergence of different local eukaryotic communities.

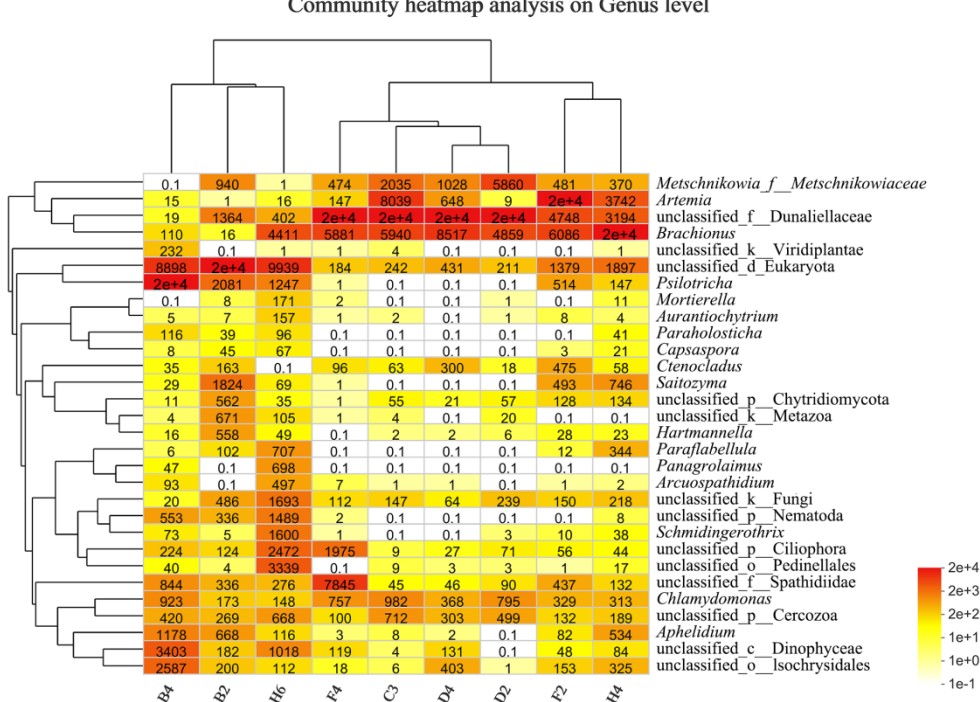

**Figure 9.** Heat map of eukaryotic taxa composition at each site (genus level) in Kyêbxang Co, Tibet, China.

### 3.2.5. Response to Environmental Factors

To intuitively reflect the relationship between environmental factors and the eukaryotic community, RDA was used to explore the correlation between the characteristics of the eukaryotic community at the phylum level and environmental factors in sediment samples from various regions of Kyêbxang Co. We found that the number of environmental factors in the study was larger than the number of samples, so we have eliminated the three environmental factors (pH, DO and Chla) with the smallest correlation coefficients based on the correlation coefficients. Table 4 shows that Depth, T, OM, N, S and C/N had significant effects on the distribution of eukaryotic communities. From Figure 10, the percentage of variance explained by the first and second axes were 52.82% and 22.09%, respectively. The profundal zone communities were more influenced by water depth, OM, N and S, while the littoral zone communities were more influenced by T. In contrast, MC and C were not significantly related to the effect of community structure.

**Table 4.** The RDA table for the environmental factors in Kyêbxang Co, Tibet, China.

|          | RDA1     | RDA2     | r2     | *p*_Values |
|----------|----------|----------|--------|-----------|
| Depth    | −0.9706  | −0.2406  | 0.8259 | 0.008     |
| T        | 0.979    | 0.2037   | 0.8172 | 0.008     |
| MC       | −0.9845  | −0.1753  | 0.408  | 0.214     |
| OM       | −0.8654  | −0.5012  | 0.7438 | 0.022     |
| N        | −0.8858  | −0.464   | 0.6788 | 0.033     |
| C        | −0.8754  | 0.4833   | 0.3149 | 0.334     |
| S        | −0.7598  | −0.6502  | 0.7164 | 0.037     |
| C/N      | 0.6046   | 0.7965   | 0.7605 | 0.01      |

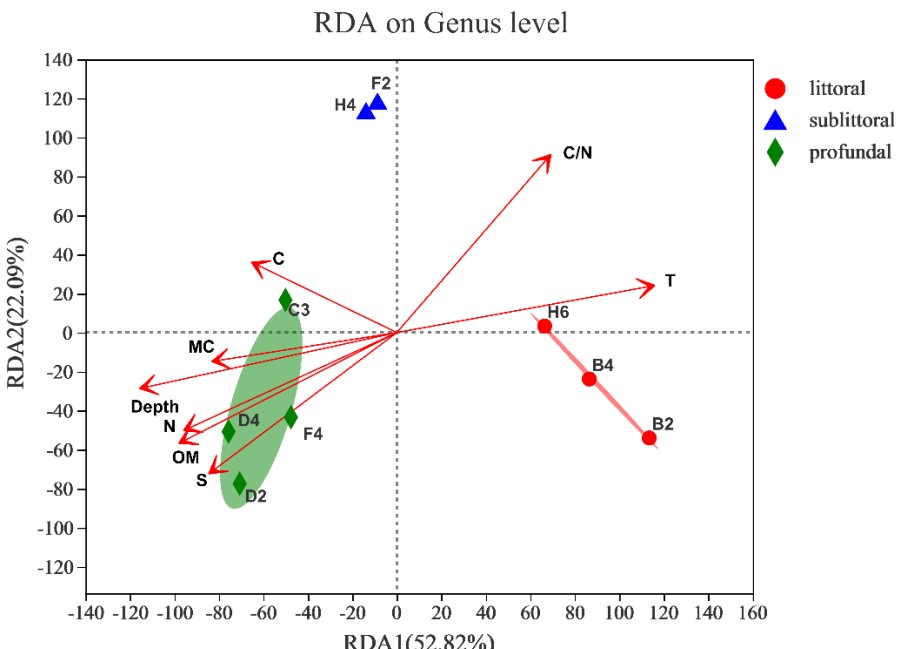

**Figure 10.** RDA ordination biplot of eukaryotic taxa and environmental factors in Kyêbxang Co, Tibet, China. The dots, triangles, and diamonds in the figure represent eukaryotes at different sites. The arrows indicate the environmental factors. The length of the arrow line indicates the correlation between the environmental factor and the sample distribution, the longer the line, the higher the correlation, and vice versa, the lower the correlation. The angle between the arrow line and the ranking axis and the angle between the arrows indicate correlation, with sharp angles indicating positive correlation and obtuse angles indicating inverse correlation. The smaller the angle, the higher the correlation.

We can determine the specific relationships between environmental factors and different taxa using a heatmap of correlation coefficients (Figure 11). For example, *Dunaliella* is a typical taxon with a significant positive correlation with depth, MC, OM, and Chla and a significant negative correlation with T. In contrast, *Psilotricha* showed a significant negative correlation with depth, MC, and OM and a significant positive correlation with T. This suggests that the two taxa live in different habitats. However, *Brachionus* did not correlate with any of these environmental factors, indicating that *Brachionus* is a planktonic organism, which stays in the water column all around the lake and affects the bottom sediments with its eggs. At the genus level, depth was associated with 14 taxa and was the absolutely dominant environmental factor affecting the abundance of the top 30 taxa. In contrast, pH was not associated with any of them. The influence of C, N, and S on organisms is generally consistent, and this trend is like the influence of water depth on eukaryotes. For example, they are positively correlated with *Dunaliella* and negatively correlated with unclassified Eukaryota. However, C is slightly different from N and S. When C is correlated with some taxa, such as Pedinellales, N and S do not correlate. However, the effect of C/N on taxa was the least significant, as it was associated with only a few taxa.

## Spearman Correlation Heatmap

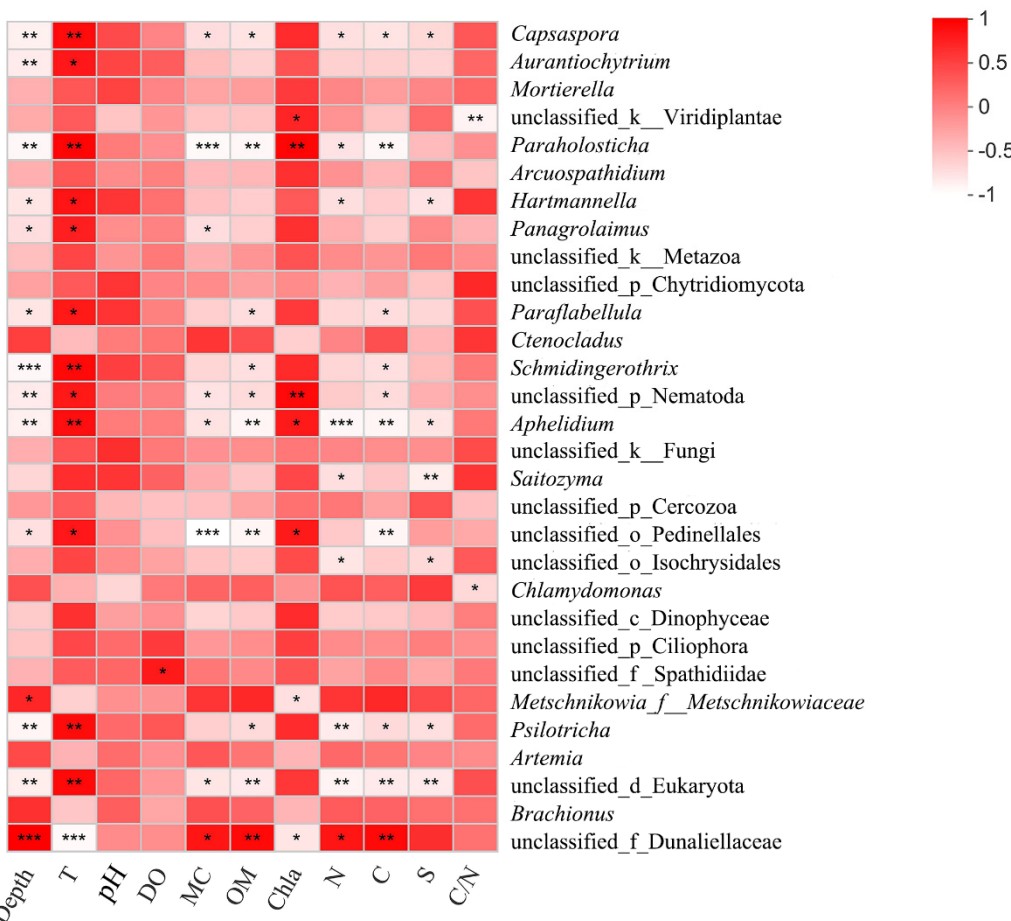

**Figure 11.** Heatmap of Spearman correlation coefficients between eukaryotic taxa and environmental factors in Kyêbxang Co, Tibet, China. Note: * indicates significant correlation at the 0.05 level; ** indicates significant correlation at the 0.01 level; *** indicates significant correlation at the 0.001 level.

### 3.2.6. Network Analysis of Eukaryotes

A single factor network analysis was conducted on the relationships between groups among the top 20 taxa in total abundance at nine sites in three locations of Kyêbxang Co, and 19 nodes and 48 edges were identified by consensus (Figure 12). Based on the two-factor network analysis of the top 30 taxa and environmental factors at the genus level, 21 nodes and 49 edges were identified by consensus (Figure 12). Circles of different colours represent different taxa; circle size represents taxonomic abundance. The green line represents negative correlation, the red line represents positive correlation, and the strength of the correlation, the thicker the line, indicates strong the correlation.

The results showed high connectivity between taxa and taxa and environmental factors in Kyêbxang Co eukaryotes. The nodes in the single factor network were divided into 11 phyla, and the values of the central coefficient for all the nodes in the network showed that Dunaliellaceae, unclassified Eukaryota, and *Aphelidium* represented important nodes throughout the network. It should be noted that Nematoda, although not highly abundant, is also significantly associated with up to 10 other organisms, indicating its importance in eukaryotic habitats. There were 21 nodes in the two-factor network analysis, of which 15 represented taxa and 6 represented environmental variables. The central coefficient values of all network nodes indicated five taxa (Dunaliellaceae, Pedinellales, *Paraholosticha*, *Aphelidium*, Nematoda) and five physicochemical factors (T, depth, MC, OM, Chla) represented the significant nodes in the network. T, like a sun, showed a positive correlation with 11 taxa, while it only showed a significant negative correlation with Dunaliellaceae.

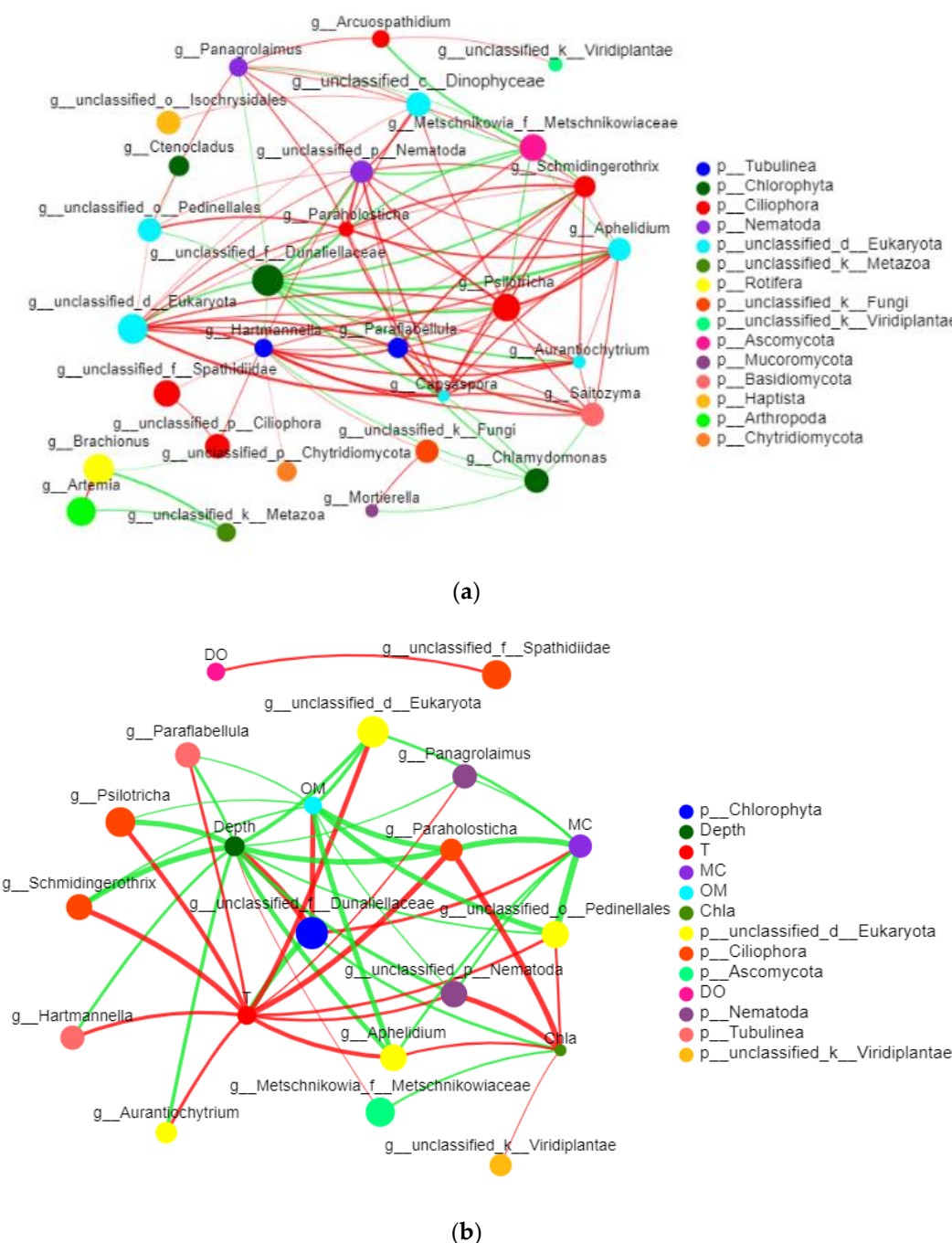

**Figure 12.** Single- and two-factor network analysis of eukaryotes in Kyêbxang Co, Tibet, China. Note: Single-factor network analysis (**a**) and two-factor network analysis (**b**). Taxa with $p < 0.05$ are shown by default. The size of the nodes in the figure indicates the abundance of taxa, and different colours indicate different taxa. The colour of the line indicates the type of correlation; red indicates a positive correlation, and green indicates a negative correlation. The thickness of the line indicates the size of the correlation coefficient; the thicker the line is, the higher the correlation between taxa. The more lines there are, the closer the connection between the nodes is.

## 4. Discussion

Benthic eukaryotes play the roles of primary producers, consumers, and decomposers in aquatic ecosystems and affect the structure of aquatic food webs from aspects of biological composition, abundance, biomass, and biodiversity [30,31]. However, few studies have been done on the composition, function, and relationship between eukaryotes and environmental factors in salt lakes [32–34]. Compared with traditional microscopic detec-

tion methods, high-throughput sequencing technology has extensively promoted the study of eukaryotic biodiversity, especially meiofauna and microfauna [6,11,35]. In this study, based on the V4 region of the 18SrRNA gene, a high-throughput sequencing method was used to find that eukaryotic diversity was high in Kyêbxang Co salt lake, Xizang Province. The nine sites sampled in Kyêbxang Co could be divided into the littoral, sublittoral, and profundal zones according to their geographical location. Eukaryotic taxa mainly include Chlorophyta, unclassified Eukaryota, Rotifera, Ciliophora, Arthropoda, Ascomycota, and other taxa.

### 4.1. Community Composition of Eukaryotes in Kyêbxang Co Salt Lake

In this study, *Dunaliella* was the dominant taxon in the profundal zone. *Dunaliella* is a taxon of Chlorophyta and is the major primary producer in many highly saline water environments, such as the Dead Sea and the Great Salt Lake [36]. *Artemia* in the salt lake can significantly feed on *Dunaliella*, thus limiting the biomass of *Dunaliella*. This study also found that *Artemia* was mainly distributed in sites in the sublittoral zone, while *Dunaliella* was the dominant taxon in the profundal zone. Therefore, the feeding action of *Artemia* may have an impact on the abundance of *Dunaliella*. In addition, the presence of so many Chlorophyta in the profundal zone may originate from the resting stages produced by phytoplankton in the upper water column [37,38]. Previous studies have found that arthropods in the profundal of Lake Ohrid are about 30% less abundant than in the littoral zone [6]. This study also found many Arthropoda and Rotifera in the sublittoral zone, but almost none in the littoral and profundal zones. The study also found that Lake Ohrid was dominated by Annelida (36%), while only one Annelida taxon was detected in Kyêbxang Co Salt Lake. Oligochaeta and Chironomidae were also dominant eukaryotic taxa in Laurentian Great Lakes [39]. This may be because Lake Ohrid and the Laurentian Great Lakes are freshwater Lakes, while Kyêbxang Co is a salt lake with a salinity above 40. 11%. Ciliophora (cilium subphylum) was also found in Lake Ohrid, with the highest abundance in the littoral zone. This is consistent with the research results of this paper. In Kyêbxang Co, Ciliophora occupied a considerable proportion in the littoral, accounting for 40.10% (B4), 8.11% (B2), and 19.51% (H6), respectively. Cercozoa, previously considered parasitic or predatory protozoa, were present at all sites. One of the significant differences between microbial eukaryotes in sediments and those in water is that Cercozoa-related OTUs are more dominant in sediments [40,41]. In a recent study, Wu and Huang reported that Cercozoa protozoan communities dominated the surface seafloor sediments of the South China Sea, accounting for 40% and 25% of the total sequence and OTUs richness, respectively [42]. Cercozoa were also found at all sites in this study, suggesting that they may play a key role in littoral and profundal sediments, which is worth exploring in more detail in future studies. Univariate network analysis showed that *Psilotricha* and *Aphelidium* were significantly negatively correlated with *Dunaliella*. *Pedinellicha* is a genus for species of Ciliophora that probably feed on *Dunaliella*. However, *Aphelidium* is a poorly known group of parasitoids of algae. The parasitoid encysts and penetrates the host alga through an infection tube. Cyst germination leads to a young trophont that phagocytes the algal cell content and progressively develops a plasmodium [43].

Compared with the mangrove intertidal zone study, it was found that the leading groups in the mangrove are copepods, diatoms, Ciliophora, and polychaetes [44]. This is generally consistent with the research results in this paper, but in the study of Wu, the dominant first producer was diatoms. Ouyang et al. also found many diatoms in the eukaryotic community in the coastal sediment of Xiamen. In contrast, diatoms accounted for a small proportion of the eukaryotes of Kyêbxang Co [45], and Chlorophyta was the dominant taxon in Kyêbxang Co Salt Lake. In addition, certain biomass of Ascomycota was also found in the study of Wu, which was also the same as the result of this study, and there was little difference in Ascomycota in the three regions of this study, indicating that Ascomycota may play a special role in forming eukaryotic groups [44]. Ascomycota occupies a dominant position in the global soil fungi community. Ascomycota, mostly soil

saprophytes, is an essential decomposer of soil organic matter and plays a vital role in the degradation of complex organic matter, promoting soil nutrient cycling and playing a necessary role in plant growth [46].

Unlike previous studies, the traditional morphological analysis showed that nematodes were the absolute dominant group of meiofauna, even up to 90%. In contrast, high-throughput sequencing showed that nematodes accounted for a low percentage, even less than 1%. Our laboratory also studied the morphological data of nematodes (unpublished data). After sorting and identification, it was found that there were 22 genera, 13 families and two classes, six orders. A total of 11 Nematode OTUs were obtained from the analysis of 18S sequencing results, among which 6 OTUs could not be identified with specific information, and the 5 OTUs identified in genus did not match the species in morphological taxonomy. Similar findings have been made in previous studies. Wilden found that the diversity of eukaryotes in deep-sea sediments is higher than that in coastal habitats, which is incomplete in contrast to earlier studies of meiofauna with identical samples, which found that the diversity in deep-sea is lower than that in coastal habitats [6]. Morphology, DNA barcoding, and metal carbonate were compared to characterize the nematode community. Only three species (13.6%) were shared among these methods [47]. Other researchers have also found no good match between molecular and traditional morphological methods [48]. This indicates that the study of community structure by the eDNA method will be affected by many factors. For example, DNA has a complex existence mechanism in water. Many environmental factors affect its generation, degradation, and movement process (such as ultraviolet, water temperature, pH, water flow, water substrate, etc.) [49]. In addition, the sampling method may have a critical impact on the experimental results. For example, morphological identification requires a larger sample of sediment than that required for DNA extraction. Only about 1 g of sediment was used for DNA extraction, and such a small sample may not contain all the species information, which would lead to such bias [50]. In addition, primers for PCR amplification also significantly influence experimental results. However, in a word, eDNA technology can provide a convenient and efficient macro description of community structure in the study area [51].

In addition, nearly 1/3 of eukaryotes in the study could not be effectively identified as species, indicating a lack of molecular data for organisms living in salt lakes in Tibet in the public database. Further research in related aspects is needed to expand the database and improve annotation information for species-level taxa.

The present study was mainly focused on the benthic eukaryotic diversity because it is inevitable that plankton will be included in the scope of the study. Plankton in the water column can enter the sediment by settling and resting stages. Plankton resting stages are a special group of organisms in sediments, and their species composition and distribution characteristics are closely related to environmental factors [52]. When the environment changes, plankton often sinks to the bottom in the form of resting stages, and when the environmental conditions are suitable, the resting stages of plankton can sprout into nutrient cells and return to the water column [53]. Resting stages of plankton were sampled in surficial sediments in the port of Haifa, Israel, which found that cysts may last in sediments much longer than in the water column [54]. Morard et al. found that the largest portion of the metabarcoded sample originated from benthic bottom-dwelling foraminifera, representing the in situ community, but a small portion (<10 %) of the metabarcodes could be unambiguously assigned to planktonic taxa, which live only in the water column [55]. In the Gulf of Finland study, calanoid copepod resting eggs were found throughout the sediment layer, with the highest abundance of resting eggs in the upper 1 cm of surface sediment—more than 10 times that of the other sediment layers, accounting for almost 70% of all resting eggs [56]. In our sampling, the sediment for DNA extraction was surface sediment. In addition, many resting eggs of Rotifera were found in our study, which is consistent with the findings of the Gulf of Finland study. Our study found that Rotifera was present in low levels at two shallow sites, B2 and B4, while they were present in more than 10% at all other sites. They even reached 59.98% at site H4. It was also found that

Chla, salinity, temperature, C and N could explain the distribution of Rotifera abundance. However, it was noted that the genus *Brachionus* of the Rotifera phylum was not correlated with any of the environmental factors (Figure 11). This may be because Rotifera resting eggs were most abundant in sediment at the 4–5 cm depth. Holm et al. [57] reported that a total of 42 copepods could produce resting eggs to survive during the cold or hot periods in their environment, with no apparent relationship with chlorophyll content. Belmonte and Rubino (2019) [58] listed 58 species of copepods that can produce resting eggs. Hundreds of other planktonic species adopt the common strategy of producing resting eggs to ensure reproduction and avoid unfavourable conditions. During the resting period, these species are in lower abundance in the water, while many stays in the sediment, waiting for favourable conditions to germinate [58]. In deeper waters of the present study, there were large numbers of Dunaliellaceae, the sole producer in many hypersaline environments, and therefore they produce many resting eggs to settle into the sediment. Moreover, because of the low abundance of benthic fauna in deep water regions, the resting eggs can maintain high numbers in sediment and lack the consumption of benthic fauna [59]. In conclusion, the existence time of plankton resting eggs in sediments is periodical [60], while most benthic eukaryotes live in sediments throughout their entire life cycle. They have weak movement ability and can respond to the environmental quality sensitively, which is more stable than plankton [61].

*4.2. Relationship between Eukaryotes and Environmental Factors*

In our study, RDA results showed that water depth, temperature, and organic matter were significant factors affecting the community distribution of eucaryotes in the benthos of Kyêbxang Co salt lake. Eukaryotes in sediments were correlated with water depth, temperature, organic matter, water content, dissolved oxygen, Chla, and other physicochemical parameters [62,63]. Previous studies have found that water depth can strongly influence eukaryotic communities in littoral and profundal sediments [42,64,65], which was also confirmed by this study. The water depth may be a critical factor in the construction of eukaryotic communities because it can substitute for many physical and chemical variables, including water temperature, salinity, pressure, nutrients, light, dissolved organic carbon, etc. [66]. The most significant environmental difference among the three locations was water depth, resulting in significant community structure differences. The Sobs index is mainly used to show the actual OTU number. Among the nine sites, the Sobs index in the littoral zone was significantly higher than in the sublittoral and profundal zones. The Shannon–Wiener index varied from 1.25 to 3.36, and the site with the highest diversity was the littoral zone. This may be because the water depth was shallow, and the light intensity was sufficient, so many plants could have enough light for photosynthesis and produce a large amount of dissolved oxygen, which facilitates the survival of other aerobic organisms. In the profundal zone, the Shannon–Wiener and Sobs index decreased significantly, which may be related to the lack of light, reduced dissolved oxygen content, and too much pressure. Very few benthos organisms can live here, but mainly the planktonic ones (as resting stages) come from the water column.

Studies have shown that salt is one of the critical factors affecting the eukaryotic community, while temperature has the most negligible impact on the community. In this study, the temperature was an essential factor influencing the eukaryotes, and many important organisms such as *Dunaliella* showed a strong positive correlation. *Dunaliella* is considered the principal or even the only primary producer in many high-salt environments [67]. The availability of inorganic nitrogen in salt lakes seems to be the main environmental factor limiting the yield of *Dunaliella* [68]. The research results in this paper also show that *Dunaliella* was significantly correlated with N content. But *Dunaliella* was less abundant in the sublittoral zone and was more than half of all organisms in the profundal zone because *Dunaliella* is a planktonic alga with a shallower water column in littoral/sublittoral than in the profundal zone. Chlorophyta includes photosynthetic autotrophs that generally need to live in the sublittoral zone with more sunlight, but the study found the opposite.

One possible explanation is that Chlorophyta are planktonic organisms and their amount in the sediment is not representative of their true abundance in the water column. Another explanation is that other large plants live in the littoral zone, and these plants compete with Chlorophyta for the ecological niche, resulting in the loss of the dominant status of Chlorophyta [69].

Dai et al. investigated eukaryotes and screened 15 eukaryotic families, mainly belonging to Cercozoa, Alveolata, and Chlorophyta, as the dominant species in this paper [70]. Among them, 13 populations were negatively correlated with temperature. Their relative abundance decreased due to temperature increase, and two were positively correlated with temperature. Therefore, the temperature does affect the community structure of eukaryotes. Burlakova analysed benthic eukaryotic community spatial gradients and temporal trends in the Lawrence Great Lakes from 1998 to 2014 [35]. Environmental factors included basicity (mg $CaCO_3$/L), chlorophyll a ($\mu$g/L), specific conductivity ($\mu$mho/cm), total dissolved phosphorus (TDP, $\mu$g/L), total phosphorus (TP, $\mu$g/L), turbidity (F/NTU) and pH. The total species richness and density of benthic eukaryotes correlated with both Chla and depth, and the species richness increased with the increase of Chla. Benthic eukaryotic density also increased with Chla concentration and decreased with depth. The negative correlation between benthic eukaryotic density and depth can be partially explained by the inverse relationship between Chla and depth, with Chla decreasing rapidly with site depth. There were significant differences in the organic matter between the three regions in our study. Still, there was no significant correlation with Chla, which may be due to the degradation of Chla due to sampling preservation. Lake productivity and depth strongly influenced community patterns, suggesting that the availability of organic carbon may be an important driver of benthic eukaryotic richness and diversity in Kyêbxang Co salt lake. Food availability, which generally decreases with increasing depth, is one of the most important factors governing the structure and function of benthic eukaryotic communities [71,72]. The dependence of deep-sea organisms on organic detritus produced in offshore surface waters, which are progressively utilized and degraded as they descend through the water column, limits deep-sea benthic eukaryotic abundance and species richness [73].

Because eukaryotes are sensitive to changes in water quality, they can be used to monitor and evaluate environmental conditions [74]. Gaedke et al. found that the community structure of microeukaryotes was closely related to freshwater nutrient levels in different lakes [75]. Chen et al. found that the genetic diversity of microplankton eukaryotes was significantly correlated with total phosphorus concentration in other Taihu Lake regions. There is a complex mutual restriction and promotion relationship between environmental factors and organisms. A detailed understanding of their corresponding relationship is conducive to maintaining and improving the ecological balance of salt lakes [76]. In recent years, high-throughput sequencing technology has been applied to analyse the composition and structure of eukaryotic taxa, and its application in biomonitoring has attracted more and more attention. High-throughput sequencing provides a suitable method for studying benthic eukaryotes, which can reveal the composition and function of benthic eukaryotes in aquatic ecosystems and provide a good reference for detecting the salt lake environment.

## 5. Conclusions

In July 2020, using Illumina Miseq high-throughput sequencing technology, 208 species of 167 genera in 120 families, 77 orders, and 37 phyla of eukaryotes were identified in the sediments of the Tibetan salt lake, Kyêbxang Co. Among them, the dominant taxon in the littoral zone was unclassified Eukaryota, the dominant taxa in the sublittoral zone were Rotifera and Arthropoda, and the dominant taxon in the profundal zone was Chlorophyta. The diversity of eukaryotic communities in the littoral, sublittoral, and profundal zones was significantly different, and the littoral zone was richer in biological types. The correlation analysis between eukaryotes and environmental factors revealed that depth, temperature, and organic matter had significant correlations with the community distribution at the level of the eukaryotic phylum. The correlation of different environmental factors with

other organisms also differed, with *Dunaliella* showing a significant positive correlation with depth and a significant negative correlation with temperature. In contrast, unclassified Eukaryota showed the exact opposite. In the co-occurrence network, the dominant taxa, *Dunaliella* and unclassified Eukaryota were at the centre of the co-occurrence network with environmental factors temperature and Depth, indicating that they are essential for the structure of the whole community. This study used high-throughput sequencing to fill the research gap of benthic eukaryotes in salt lakes and provide more references and new ideas for subsequent studies.

**Author Contributions:** L.H. completed the experiment, data analysis, and wrote the draft manuscript. Q.W. completed the field sampling and experiment. Z.W. contributed to data collection and processing. F.W. carried out the field sampling and data collection. S.S. gave advice and guidance to the field sampling and manuscript revision. X.L. carried out the field sampling, advised the data analysis and contributed to the manuscript revision. All authors have read and agreed to the published version of the manuscript.

**Funding:** This research was funded by the Fundamental Research Funds for the Central Universities: 201964024 and the Science and Technology Project of the Tibet Autonomous Region: XZ201703-GB-04 and XZ202102YD0022C.

**Data Availability Statement:** The Sequence Data used in this study can be accessed in the Sequence Read Archive (SRA) database of the NCBI with accession number PRJNA818805.

**Acknowledgments:** We are grateful to the assistance provided by Long Hongan, Zhu Boshan and Wang Pengfei in the field sampling in Tibet, China.

**Conflicts of Interest:** The authors declare that the research was conducted in the absence of any commercial or financial relationships that could be construed as a potential conflict of interest.

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
