# Peer review of "Eukaryotic Diversity Based on High-Throughput 18S rRNA Sequencing and Its Relationship with Environmental Factors in a Salt Lake in Tibet, China"

_water, doi:10.3390/w14172724_

Round 1

Reviewer 1 Report

In this paper, authors investigations the benthic eukaryote compositions in the Tibetan salty lake by using high-throughput sequencing technology. And the environmental factors on the compositions were clarified, which introduced the benthic eukaryotes in plateau salty lake. In my opinion, the differences compared with the plain area were the expected discussions for readers.

 1. It is too wordy to use a whole paragraph in the introduction to introduce high-throughput sequencing technology. After all, this technology has been widely used. Please simplify them.

2. Line 126, 132, 135, …: p must be italicized, and it should be consistent in uppercase or lowercase letter.

3. Table 1: Water temperature varied significantly at the studied sample sites. Please verify or explain them.

4. Line 185-186: “Nine samples were sequenced by Illumina miseq platform in Shanghai Meiji Biomedical Technology Co., Ltd.” This sentence must be moved to methods part.

5. Table 3: Please provide standard deviation.

6. Figure 9, 12, …: Scientific name should be italicized.

7. Line 392-400, Line 416-419: This part belongs to the figure or table illustration, not the result.

8. Line 516-529: Whether microscopic examinations of algal composition of these samples were performed, so as to compare with sequencing results. Algal compositions determined by sequencing technology were always different from microscopy results. And the results of the references were identified by DNA sequences?

9. Line 533: It seemed that it is not advised to discuss a lot by using unpublished data.

10. Line 572: OUT -> OTU, and “Sobs index is mainly used to show the actual OTU number.” This sentence must be move to methods part.

11. Line 603: CaCO3

12. Line 91, Line 603: “chlorophyll a” must be consistent in the text.

Reviewer 2 Report

By He et al.,

Benthic eukaryotic diversity based on high-throughput 18S rRNA sequencing and its relationship with environmental factors in a salt lake in Tibet, China

Major comment:

We have a little understanding for the community structure and diversity for Eukaryote in various environments. Most studies are focused on those of Prokaryote. Recently, based on the metagenomic study, three domain concept is converted into two domain. Nonetheless, still we should focus on their ecological functions. Viewed in this light, this study might provide to understand for eukaryotic diversity in salt lakes. Before considering publication, the authors should be considering some comments.

Minor comment:

L23, provide more genera

L34-36, provide some examples.

L61, typo, Artemia

L67, should find more suitable terminology, on behalf of the “enrich”.

L87, should be replaced by characterizations? Indices??

L103, eukV4r?  need reference (AEM) for the primer set.

L113, for diversity analysis, which program used; mothur or qiime?

L180, table 1. OM and C are different value?

L234-235, this mention seems to be not correlated with figure 2. The graph of figure 2 did not meet the plateau.

L251, figure 4. In this study, student’s t-test is inappropriate statistics. You should use other statistics methods such as Mann-Whitney U test.

L389, develop the quality for figure 9. Also, the value of heat should be indicated.

L414, figure 10, please provide more information for (a) and (b). which were differences between them?

L434, figure 11 need more clarity.
